# The Neutrophil-Lymphocyte Ratio as an Early Predictive Marker of the Severity of Post-Endoscopic Retrograde Cholangiopancreatography Pancreatitis

**DOI:** 10.3390/medicina58010013

**Published:** 2021-12-22

**Authors:** Sang Hoon Lee, Tae Yoon Lee, Young Koog Cheon

**Affiliations:** Department of Internal Medicine, Konkuk University School of Medicine, Seoul 05030, Korea; lshjjang_2000@hanmail.net (S.H.L.); yksky001@hanmail.net (Y.K.C.)

**Keywords:** endoscopic retrograde pancreatography, pancreatitis, prognosis, neutrophil–lymphocyte ratio

## Abstract

*Background and objectives:* Factors predictive of severe non-iatrogenic acute pancreatitis have been investigated, but few studies have evaluated prognostic markers of severe post-endoscopic retrograde cholangiopancreatography (ERCP) pancreatitis (PEP). The neutrophil–lymphocyte ratio (NLR) has been studied for predicting severe acute pancreatitis. We examined the predictive value of NLR in patients with PEP. *Materials and Methods:* From January 2012 to August 2021, 125 patients who developed PEP were retrospectively evaluated. The NLR was measured before, and on days 1 and 2 after, ERCP. PEP was categorized as mild, moderate, or severe according to consensus guidelines, based on the prolongation of planned hospitalization. Patients were divided into two groups, mild-to-moderate vs. severe PEP. *Results:* We analyzed 125 patients with PEP, 18 (14.4%) of whom developed severe PEP. The baseline NLR was similar between the two groups (2.26 vs. 3.34, *p* = 0.499). The severe PEP group had a higher NLR than the mild/moderate PEP group on days 1 (11.19 vs. 6.58, *p* = 0.001) and 2 (15.68 vs. 5.32, *p* < 0.001) post-ERCP. The area under the curve of the NLR on days 1 and 2 post-ERCP for severe PEP was 0.75 (95% confidence interval (CI), 0.64–0.86)) and 0.89 (95% CI, 0.81–0.97), respectively; NLR on day 2 had greater power to predict severe PEP. The optimal cutoff value of the NLR on days 1 and 2 after ERCP for prediction of severe PEP was 7.38 (sensitivity, 72%; specificity, 69%) and 8.17 (sensitivity, 83%; specificity, 83%), respectively. In a multivariate analysis, a Bedside Index of Severity in Acute Pancreatitis score ≥3 (odds ratio (OR) 9.07, *p* = 0.012) and NLR on day 2 > 8.17 (OR 18.29, *p* < 0.001) were significantly associated with severe PEP. *Conclusions:* The NLR on day 2 post-ERCP is a reliable prognostic marker of severe PEP.

## 1. Introduction

Acute pancreatitis is the most serious adverse event associated with endoscopic retrograde cholangiopancreatography (ERCP). The incidence of post-ERCP pancreatitis (PEP) ranges from 4% to 15% [1,2,3]. Although the incidence of severe PEP is low (0.3–0.6%) [4], it is associated with high morbidity, prolonged hospitalization and, occasionally, mortality. The mortality rate of PEP is rising [5]. This necessitates prediction of severity as early as possible.

Although risk factors for PEP have been investigated, few studies have focused on predictors of PEP severity. Suggested prognostic markers of PEP include serum phosphate and systemic inflammatory response syndrome (SIRS) [6,7]. There is a need for new, early, and simple predictors of severe PEP, to enable stratification of patients for close monitoring, vigorous hydration, and selective transfer to the intensive care unit.

The neutrophil–lymphocyte ratio (NLR) is reportedly predictive of adverse events in acute pancreatitis unrelated to ERCP [8,9,10,11]. The NLR reflects the inflammatory and immune response more accurately than the total white blood cell (WBC) count [12]. The NLR is predictive of disease severity, organ failure, local complications, and intensive care unit admission in acute pancreatitis [8,10]. However, no study has evaluated the association of the NLR with severe PEP. We examined the utility of the NLR for predicting the severity of PEP.

## 2. Materials and Methods

### 2.1. Patients and Data Collection

We conducted a retrospective study of patients who underwent ERCP between 1 January 2012 and 31 August 2021 at Konkuk University Medical Center, Seoul, Korea. A prospectively collected ERCP database of 4189 patients was reviewed. Four experienced endoscopists performed all of the ERCPs. Demographic (age, sex, and underlying diagnosis), clinical, and endoscopic data were obtained from the electronic medical records. The clinical data comprised laboratory values (hematocrit, WBC count, NLR, C-reactive protein (CRP), glucose, calcium, and blood urea nitrogen (BUN)), which were collected before and 1 and 2 days after ERCP. The endoscopic data comprised the goal (diagnostic vs. therapeutic) of ERCP and placement of a pancreatic stent. A prophylactic pancreatic stent was placed only in case of repeated unintentional cannulation of the pancreatic duct during ERCP. Vital signs (body temperature, blood pressure, heart rate, and respiratory rate) were measured immediately and every 6–8 h after ERCP. We also evaluated SIRS based on the vital signs and WBC count within 48 h after ERCP.

For the prevention of PEP, patients who underwent ERCP received protease inhibitors (e.g., gabexate mesylate, nafamostat, ulinastatin) and aggressive intravenous hydration (a 10–20 mL/kg bolus before the ERCP, 3 mL/kg/h during the ERCP, and 3 mL/kg/h after the ERCP) with lactated Ringer’s solution. The diagnosis of PEP was made 6–24 h after ERCP. If a patient was suspected of having PEP, administration of protease inhibitors and the rate of fluid therapy remained until 48 h after ERCP. Rectal non-steroidal anti-inflammatory drugs (NSAIDs) were not used because those were not available in Korea.

We included patients subsequently diagnosed with PEP after ERCP. We excluded patients who had concurrent acute inflammatory diseases (e.g., acute cholangitis and cholecystitis, liver abscess, pneumonia) at admission, because these can confound the association between the NLR and PEP. We excluded patients <18 years of age, those who experienced other adverse events associated with ERCP (e.g., perforation, bleeding, and biloma), and those with missing data. This study was approved by the Institutional Review Board of the hospital. Informed consent was waived by the board. 

### 2.2. Study Outcomes and Definitions

The primary outcome was whether the NLR on day 2 post-ERCP could predict the severity of PEP. The secondary outcome was the optimal NLR cutoff value for predicting severe PEP.

PEP was defined as the presence of the following: characteristic acute abdominal pain lasting for at least 24 h after ERCP with serum amylase and/or lipase level ≥ 3-fold higher than the upper limit of normal; and/or abdominal computed tomography consistent with acute pancreatitis. PEP was categorized as mild, moderate, or severe according to consensus guidelines: mild (PEP requiring hospitalization for 2–3 days); moderate (PEP requiring hospitalization of 4–10 days), and severe (PEP requiring hospitalization for more than 10 days and/or the presence of any of the following: acute fluid collections, pancreatic necrosis, persistent organ failure, surgical or percutaneous intervention, or mortality) [13]. Patients with PEP were classified into mild-to-moderate and severe PEP groups, and the association between the NLR and PEP severity was investigated. The NLR was determined by calculating the ratio between the absolute neutrophil and lymphocyte counts, and NLR values were analyzed before ERCP and on day 1 and day 2 after ERCP. SIRS was defined as fulfilling at least of two of the following criteria: body temperature >38 °C or <36 °C, heart rate >90 beats/min, respiratory rate >20 breaths/minute, WBC count >12,000 or <4000/µL, and > 10% immature bands.

### 2.3. Statistical Analysis

We used SPSS for Windows 25.0 software (SPSS Inc., Chicago, IL, USA) for the statistical analysis. Descriptive statistics for continuous and categorical variables are presented as means ± standard deviation and numbers (%), respectively. Differences between categorical variables were analyzed using the chi-square test and Fisher’s exact test, and those between continuous variables using Student’s *t*-test or Mann–Whitney U test, as appropriate. When continuous variables were not normally distributed, they were described as medians with interquartile range and the Mann–Whitney U test was used to compare them. A receiver operating characteristic (ROC) curve was plotted and the area under the curve (AUC) was calculated to determine the optimal cutoff NLR value; 95% confidence intervals (CI) were also obtained. To determine the independent predictive power of the NLR for severe PEP, multivariate regression analysis was performed, using the variables with *p*-value < 0.15 in the univariate analysis. To prevent the problem of multicollinearity due to significant correlation between WBC count, SIRS, and Bedside Index of Severity in Acute Pancreatitis (BISAP) score, we selected the BISAP score for multivariate analysis. A *p*-value < 0.05 indicated statistical significance.

## 3. Results

Among 4189 patients who underwent ERCP over 9 years, 170 (4.0%) were diagnosed with PEP. After excluding 45 patients, 125 (3.0%) satisfied the inclusion criteria and were analyzed (Figure 1). 

Of these 125 PEP patients, 83 (66.4%) had mild, 24 (19.2%) had moderate, and 18 (14.4%) had severe PEP. Among the 18 patients with severe PEP, 17 had a hospitalization duration >10 days, 7 developed pancreatic necrosis, 4 underwent percutaneous drain insertion, 1 underwent surgery, and 1 died. The demographic and clinical features of the patients are listed in Table 1.

The severe PEP group (*n* = 18) was not significantly different from the mild/moderate PEP group (*n* =107) in age, sex, indications for ERCP, goal of ERCP, use of a prophylactic pancreatic stent, or hematocrit, CRP, glucose, BUN, calcium, and albumin levels. Patients with severe PEP had a higher BISAP score (2.44 vs. 1.07, *p* < 0.001) and WBC count (11,910 vs. 8050, *p* = 0.004) on day 1 post-ERCP. The incidence of SIRS within 48 h after ERCP was significantly higher in the severe PEP group (77.8% vs. 21.5%, *p* < 0.001). The hospitalization duration in the severe PEP group was longer than in the mild/moderate PEP group (23.39 vs. 8.84 days, *p* < 0.001).

The NLR values in the mild/moderate and severe PEP groups are listed in Table 2. Although the NLR before ERCP was similar between the two groups (2.26 vs. 3.34, *p* = 0.499), the severe PEP group had a significantly higher NLR than the mild/moderate PEP group on days 1 (11.19 vs. 6.58, *p* = 0.001) and 2 (15.68 vs. 5.32, *p* < 0.001) post-ERCP.

In the mild/moderate PEP group, the NLR peaked on day 1 post-ERCP and decreased on day 2, indicating resolution of inflammation. However, in the severe PEP group, the NLR continued to increase on day 2 post-ERCP, indicating significant ongoing inflammation.

The ROC curves for the NLR as a predictor of severe PEP are shown in Figure 2. The AUC was 0.75 (95% CI, 0.64–0.86) and 0.89 (95% CI, 0.81–0.97) for NLR on day 1 and day 2, respectively; therefore, NLR on day 2 had greater power to predict severe PEP. The optimal cutoff values of NLR on days 1 and 2 for predicting severe PEP were 7.38 (sensitivity, 72%; specificity, 69%) and 8.17 (sensitivity, 83%; specificity, 83%), respectively. Two other relevant cutoff points for NLR on day 2 were 9.23 (sensitivity, 72%; specificity, 89%) and 15.64 (sensitivity, 61%; specificity, 98%), respectively.

In univariate logistic analyses, a BISAP score ≥3, leukocytosis, SIRS, NLR on day 1 >7.38 and NLR on day 2 >8.17 were significantly associated with PEP severity (Table 3).

A multivariate analysis controlling for related variables showed that a BISAP score ≥3 (odds ratio [OR] 9.07, *p* = 0.012) and NLR on day 2 >8.17 (OR 18.29, *p* < 0.001) were significant predictors of severe PEP (Table 4).

## 4. Discussion

We investigated whether the NLR can predict severe PEP. We found that the NLR can discriminate patients who develop mild/moderate and severe PEP. The NLR reflects a dynamic process, with the optimal cutoff value thereof for predicting PEP severity changing over time after onset, moving toward normal in those with mild/moderate PEP and remaining elevated in those with severe PEP. Moreover, NLR on day 2 post-ERCP had the highest predictive power (AUC = 0.89) for severe PEP among the variables tested. A cutoff value of 8.17 showed a sensitivity of 83% and specificity of 83% for predicting severe PEP. A BISAP score ≥3 was also significantly associated with severe PEP.

Since Zahorec et al. first reported NLR as a reliable predictor of systemic inflammation in critically ill patients in 2001 [14], it has been investigated in malignant and benign diseases [15,16,17]. The higher the NLR, the more imbalanced the inflammatory status, because the NLR reflects the balance of neutrophilia (representing systemic inflammation) and lymphopenia (stress-induced cortisol response) [17]. Neutrophils promote inflammation by secreting proteolytic enzymes and proinflammatory cytokines (IL-1, IL-6) [10], leading to SIRS and organ failure in severe acute pancreatitis. Lymphocytes regulate systemic inflammation as the disease progresses. Ongoing inflammation causes lymphopenia as a consequence of lymphocyte redistribution and apoptosis. Therefore, the NLR may reflect dynamic changes in the immune system in acute pancreatitis.

The NLR is associated with the severity of acute pancreatitis [8,9,10,11,18]. Suppiah et al. reported that the NLR during the first 48 h is an independent predictor of severe acute pancreatitis [9]. Similarly, a Korean study of 490 patients reported that the NLR is associated not only with severe acute pancreatitis, but also with organ failure [8]. Wang et al. analyzed 110 patients with hypertriglyceridemia-induced acute pancreatitis, of whom 10 had severe disease. The NLR was significantly predictive of severe acute pancreatitis; the optimal cutoff value was 10 [10]. A recent meta-analysis of 10 studies reported overall sensitivity, specificity, and AUC values of 79%, 71%, and 0.82 (0.78–0.85), respectively, [11], indicating that the NLR has moderately high predictive power for severe acute pancreatitis.

PEP is the most common complication of ERCP and differs from simple acute pancreatitis in its iatrogenicity. PEP commonly influences malpractice claims related to ERCP and can be stressful for endoscopists. Even if PEP is detected in a timely fashion, the inability to predict its severity at an early stage can prevent stratification by severity and early initiation of appropriate treatment. Suggested predictors of PEP include SIRS and serum phosphate. However, a diagnosis of SIRS requires analysis of vital signs, which change over time and differ by mode of acquisition. SIRS showed moderate predictive power (AUC 0.74) for PEP severity [7]. Serum phosphate did not show satisfactory predictive power (AUC 0.65) for severe PEP [6], and is not routinely measured in all institutions. Thus, there is a need for an indicator to predict the severity of PEP with high accuracy.

NLR on day 2 after ERCP can be used to identify patients who need close monitoring, vigorous hydration, abdominal imaging, and possible transfer to the intensive care unit. Unlike prognostic markers such as BISAP and SIRS, NLR is simple to calculate, routinely measured in clinical practice, and unaffected by the examiner. Although NLR on day 1 and NLR on day 2 post-ERCP had reasonable predictive power for severe PEP in this study, NLR on day 2 had greater predictive ability than NLR on day 1 (AUC 0.89 vs. 0.75). This may be because the post-ERCP inflammatory cascade peaks 24–48 h after ERCP. Serum levels of inflammatory cytokines, including IL-6 and CRP, in severe PEP reportedly peak at 24–48 h [19]. Interestingly, SIRS at 24–48 h after ERCP is more strongly correlated with severe PEP than SIRS at 0–24 h (AUC 0.74 vs. 0.51) [7], because SIRS is mediated by inflammatory cytokines. NLR on day 2 post-ERCP may correlate with the peak levels of inflammatory cytokines seen in PEP. Thus, NLR on day 2 is a better predictor of severe PEP than NLR on day 1. The NLR in our severe PEP group increased until 2 days after ERCP, indicating ongoing inflammation. The NLR in the mild/moderate PEP group peaked on day 1 post-ERCP and was lower on day 2. The change in NLR over time can also be used to predict severe PEP.

This is the first report of the usefulness of NLR for predicting PEP severity. However, this was a retrospective study with a relatively small number of patients. Furthermore, a sample size of 122 patients in each group will be required to achieve power of 80% with a confidence level of 5% to confirm these results when we assumed a 7.5% incidence rate of mild/moderate PEP and a 0.5% incidence of severe PEP. So the study is underpowered in terms of drawing definitive conclusions on the association between the NLR and severe PEP. Additionally, the results may not generalize to patients with PEP and concurrent acute inflammatory disease or other ERCP complications, because the lymphocyte and neutrophil counts may be confounded by these conditions. Finally, we did not assess the change in NLR beyond 48 h after ERCP because most patients with mild PEP were discharged early.

## 5. Conclusions

In conclusion, NLR on day 2 post-ERCP has high discriminatory ability for severe PEP, with an optimal cutoff value of 8.17. This finding needs to be confirmed in a large prospective trial.

## Figures and Tables

**Figure 1 medicina-58-00013-f001:**
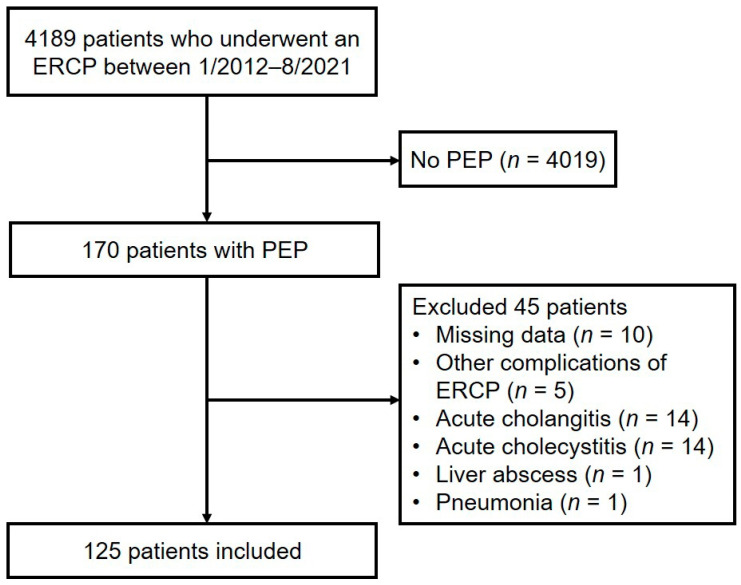
Flowchart of patient enrollment. ERCP, endoscopic retrograde cholangiopancreatography; PEP, post-ERCP pancreatitis.

**Figure 2 medicina-58-00013-f002:**
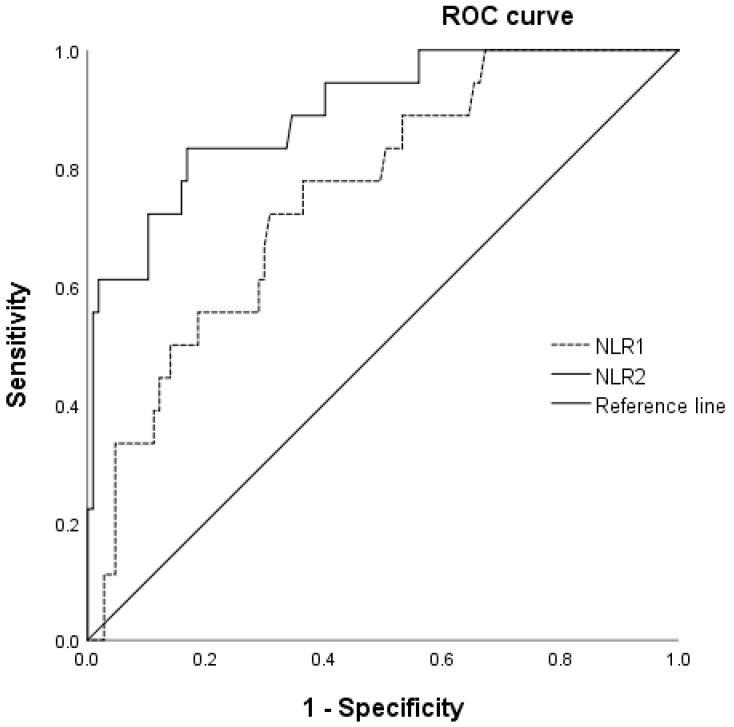
Receiver operating characteristic curve for NLR as a predictor of severe post-ERCP pancreatitis. ERCP, endoscopic retrograde cholangiopancreatography; NLR 1, neutrophil–lymphocyte ratio on day 1 post-ERCP; NLR 2, neutrophil–lymphocyte ratio on day 2 post-ERCP.

**Table 1 medicina-58-00013-t001:** Demographic and clinical characteristics of 125 patients diagnosed with post-ERCP pancreatitis.

	Mild-to-Moderate Post-ERCP Pancreatitis (*n* = 107)	Severe Post-ERCP Pancreatitis (*n* = 18)	*p*-Value
Age, mean ± SD, years	67.07 ± 13.62	60.94 ± 15.01	0.084
Male, gender, *n* (%)	52 (48.6%)	8 (44.4%)	0.803
Indication for ERCP			
Benign	70 (65.4%)	15 (83.3%)	0.13
Malignancy	37 (34.6%)	3 (16.7%)	
Purpose of ERCP			
Diagnostic	2 (1.9%)	0 (0%)	1.000
Therapeutic	105 (98.1%)	18 (100%)	
BISAP score (on the first post-ERCP day)	1.07 ± 0.58	2.44 ± 0.61	<0.001
SIRS within 48 h, *n* (%)	23 (21.5%)	14 (77.8%)	<0.001
Prophylactic pancreatic stent	12 (11.2%)	0 (0%)	0.212
Laboratory data (on the first post-ERCP day)			
Hematocrit, %	36.31 ± 4.91	37.75 ± 4.25	0.245
WBC count (×10^3^/µL)	8.05 ± 3.01	11.91 ± 4.84	0.004
CRP (mg/dL)	1.75 ± 2.13	2.91 ± 3.13	0.05
Glucose (mg/dL)	124.39 ± 43.74	107.39 ± 19.88	0.108
BUN (mg/dL)	15.2 ± 6.88	12 ± 3.96	0.058
Calcium (mg/dL)	8.85 ± 0.46	8.82 ± 0.51	0.784
Albumin (g/dL)	3.65 ± 0.39	3.5 ± 0.49	0.149
Hospital stay (d)	8.84 ± 3.74	23.39 ± 16.04	0.001

SD, standard deviation; ERCP, endoscopic retrograde cholangiopancreatography; BISAP, Bedside Index of Severity in Acute Pancreatitis; SIRS, systemic inflammatory syndrome; WBC, white blood cell; CRP, C-reactive protein; BUN, blood urea nitrogen.

**Table 2 medicina-58-00013-t002:** Changes over time in the neutrophil–lymphocyte ratio in the mild/moderate and severe PEP groups.

Neutrophil–Lymphocyte Ratio	Mild/Moderate PEP (*n* = 107)	Severe PEP (*n* = 18)	*p*-Value
NLR before ERCP	3.34 (1.68)	2.26 (1.49)	0.499
NLR on day 1	6.58 (5.21)	11.19 (10.95)	0.001
NLR on day 2	5.32 (4.98)	15.68 (13.06)	<0.001

PEP, post-ERCP pancreatitis; NLR, neutrophil–lymphocyte ratio. Values are median NLR (interquartile range).

**Table 3 medicina-58-00013-t003:** Simple logistic regression of factors predicting severe post-ERCP pancreatitis.

Variable	Odds Ratio (95% CI)	*p*-Value
Malignancy	0.37 (0.1–1.39)	0.175
BISAP score ≥ 3 points	10.2 (2.7–38.52)	0.001
WBC count > 10,000/µL	8.1 (2.63–24.87)	<0.001
CRP > 1 mg/dL	2.96 (0.98–8.89)	0.073
BUN > 24 mg/dL	0.84 (0.78–0.91)	0.388
Glucose > 126 mg/dL	0.51 (0.13–1.9)	0.396
Systemic inflammatory response syndrome	12.78 (3.83–42.5)	<0.001
NLR on day 1 > 7.38	5.58 (1.84–16.91)	0.002
NLR on day 2 > 8.17	24.72 (6.47–94.3)	<0.001

ERCP, endoscopic retrograde cholangiopancreatography; BISAP, Bedside Index of Severity in Acute Pancreatitis; WBC, white blood cell; CRP, C-reactive protein; BUN, blood urea nitrogen; NLR, neutrophil–lymphocyte ratio.

**Table 4 medicina-58-00013-t004:** Multiple logistic regression of severe post-ERCP pancreatitis.

Variable	Odds Ratio (95% CI)	*p*-Value
BISAP score ≥ 3 points	9.07 (1.61–50.99)	0.012
CRP > 1 mg/dL	2.47 (0.42–14.28)	0.312
NLR on day 1 > 7.38	0.87 (0.13–5.67)	0.884
NLR on day 2 > 8.17	18.29 (4.5–73.5)	<0.001

ERCP, endoscopic retrograde cholangiopancreatography; BISAP, Bedside Index of Severity in Acute Pancreatitis; CRP, C-reactive protein; NLR, neutrophil–lymphocyte ratio.

## Data Availability

The datasets generated for this study are available on request to the corresponding author.

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
