# Peer review of "The Neutrophil-Lymphocyte Ratio as an Early Predictive Marker of the Severity of Post-Endoscopic Retrograde Cholangiopancreatography Pancreatitis"

_medicina, 2021, doi:10.3390/medicina58010013_

Round 1
Reviewer 1 Report
Dear Editor
Thanks for inviting me to review the manuscript medicina-1505713 entitled "The Neutrophil-lymphocyte Ratio as an Early Predictive Marker of the Severity of Post-endoscopic Retrograde Cholangiopancreatography Pancreatitis".
I appreciate the authors' work to collect these data and to wrie such a good manuscript. Although there are some limitations regarding this study such as limited sample size and retrospective data collection, the title is interesting and the data are reportable. I have some recommendations to improve this manuscript.
1- The methods section of the abstract needs further details such as the PEP severity meaning and grouping the participants.
2- In line 22 of the abstract, the "NLR 2" is not clear for the readers. I suggest using the NLR on day 2 instead.
3- The methods section of the manuscript also needs further details regarding the grouping of the patients according to the PEP severity.
4- The authors need to provide further information regarding their multivariable (multivariate) analyses. Which analyses with which methods were used?
5- Did you check the normality in the distribution of the quantitative data? Please mention the test or method. Because you reported the IQR for NLR in the table two. Which means this data cannot be compared between the two groups using independent-samples Student’s t-test.
6- In the table 2, the NLR 0, 1, and 2 are not clear enough. I suggest writing the neutrophil-lymphocyte ratio on the top, so you can write three columns of Day one, Day two, and Day three instead of NLR 0, 1, 2.
7- The figure two is not proper enough for reporting a quantitative data of two groups in three phases. Instead, you can use a graph with an error bar (showing both mean and 95%CI). Or, you could remove this figure since this data is mentioned in both table two and the text (repeated results).
8- I suggest using percent sign (%) to report the specificity and sensitivity instead of point (for example 72% instead of 0.72). Please use this format for all of the manuscript.
9- Although the mentioned cut-off point is the best one for the NLR2, I think there are two other reportable cut-off points for it. You can see them in the ROC curve. They are important because of their specificities (one of them more than 85% and one more than 95%!). I strongly recommend reporting these two cut-off points too.
10- Again in discussion, the NLR 1 and 2 are not clear for the readers. Please use NLR on day 1/2 instead.
Author Response
Dear Editor
Thanks for inviting me to review the manuscript medicina-1505713 entitled "The Neutrophil-lymphocyte Ratio as an Early Predictive Marker of the Severity of Post-endoscopic Retrograde Cholangiopancreatography Pancreatitis".
I appreciate the authors' work to collect these data and to write such a good manuscript. Although there are some limitations regarding this study such as limited sample size and retrospective data collection, the title is interesting and the data are reportable. I have some recommendations to improve this manuscript.
A) Thank you very much for reviewing this manuscript. We have read your comments carefully and tried our best to address them one by one. We hope that the manuscript has been improved after this revision.
1- The methods section of the abstract needs further details such as the PEP severity meaning and grouping the participants.
A) Thank you very much for your comments. As the abstract is limited to 300 words, we have added a brief description regarding the PEP severity definition and grouping of the participants to the methods section of the abstract as follows:
“PEP was categorized as mild, moderate, or severe according to consensus guidelines, based on the prolongation of planned hospitalization. Patients were divided into two groups, mild-to-moderate vs. severe PEP.”
2- In line 22 of the abstract, the "NLR 2" is not clear for the readers. I suggest using the NLR on day 2 instead.
A) Thanks for the comment. We have replaced “NLR 2” with “NLR on day 2” in the abstract.
3- The methods section of the manuscript also needs further details regarding the grouping of the patients according to the PEP severity.
A) Thanks for your comment. We have provided further details regarding the grouping of the patients according to the PEP severity at the Method section as follows:
“PEP was categorized as mild, moderate, or severe according to consensus guidelines: mild (PEP requiring hospitalization for 2-3 days); moderate (PEP requiring hospitalization of 4–10 days), and severe (PEP requiring hospitalization for more than 10 days and/or the presence of any of the following: acute fluid collections, pancreatic necrosis, persistent organ failure, surgical or percutaneous intervention, or mortality). Patients with PEP were classified into mild-to-moderate and severe PEP groups, and the association between the NLR and PEP severity was investigated.”
4- The authors need to provide further information regarding their multivariable (multivariate) analyses. Which analyses with which methods were used?
A) Thank you for your comment for the statistical methodology. We have described in detail as follows:
“To determine the independent predictive power of the NLR for severe PEP, multivariate regression analysis was performed, using the variables with P-value < 0.15 in the univariate analysis. To prevent the problem of multicollinearity due to significant correlation between WBC count, SIRS, and Bedside Index of Severity in Acute Pancreatitis (BISAP) score, we selected BISAP score for multivariate analysis. A P-value < 0.05 indicated statistical significance.”
5- Did you check the normality in the distribution of the quantitative data? Please mention the test or method. Because you reported the IQR for NLR in the table two. Which means this data cannot be compared between the two groups using independent-samples Student’s t-test.
A) Thank you very much for pointing this out. We checked the normality in the distribution of the quantitative data with Kolmogorov-Smirnov test. We used Student’s t-test to compare continuous variables with a normal distribution. When continuous variables were not normally distributed, they were described as medians with interquartile range, and Mann-Whitney U test was used to compare them. Because NLR was not normally distributed, we used Mann-Whitney U test and described NLR as medians with interquartile range. We have revised the statistical analysis section as follows:
“Differences between categorical variables were analyzed using chi-square test, and Fisher’s exact test and those between continuous variables using Student’s t-test or Mann-Whitney U test, as appropriate. When continuous variables were not normally distributed, they were described as medians with interquartile range and Mann-Whitney U test was used to compare them.”
6- In the table 2, the NLR 0, 1, and 2 are not clear enough. I suggest writing the neutrophil-lymphocyte ratio on the top, so you can write three columns of Day one, Day two, and Day three instead of NLR 0, 1, 2.
A) Thank you for your nice recommendation. We have modified the table 2 according to your recommendation.
7- The figure two is not proper enough for reporting a quantitative data of two groups in three phases. Instead, you can use a graph with an error bar (showing both mean and 95%CI). Or, you could remove this figure since this data is mentioned in both table two and the text (repeated results).
A) Thank you very much for pointing this out. We have removed the figure 2 because repeated results are presented in both table two and the text.
8- I suggest using percent sign (%) to report the specificity and sensitivity instead of point (for example 72% instead of 0.72). Please use this format for all of the manuscript.
A) Thanks for the comment. We have replaced the point with the percent sign (%) to report the specificity and sensitivity in all of the manuscripts.
9- Although the mentioned cut-off point is the best one for the NLR2, I think there are two other reportable cut-off points for it. You can see them in the ROC curve. They are important because of their specificities (one of them more than 85% and one more than 95%!). I strongly recommend reporting these two cut-off points too.
A) Thank you very much for pointing this out. We have found two reportable cut-off points in the ROC curve. And this finding is added to as follows in lines 163-165 as follows: Two other relevant cut-off points for NLR on day 2 were 9.23 (sensitivity, 72%; specificity, 89%) and 15.64 (sensitivity, 61%; specificity, 98%), respectively.
10- Again in discussion, the NLR 1 and 2 are not clear for the readers. Please use NLR on day 1/2 instead.
A) Thank you for the nice reminder. We have used NLR on days 1 and 2 in the discussion section.
Reviewer 2 Report
Thank you for this retrospective study on the value of NLR as a predictor of severe PEP.
The centre has performed a large number of ERCPs over 10 years and have a roving average of PEP over 10 years of around 0.4% which is very good (4% over 10 years). These are within the range of international reports.
The statistics have been applied appropriately, though the calculation of power of study is missing. The discussion could mention what would be the appropriate sample size for adequate powering of the study that the authors mention was not met.
My primary concern is that there is no statement regarding the ethics of the study, even if it was a retrospective study, it is good practice to at least mention that the ethical concerns were considered e.g. patient confidentiality and consent; even if the medical records are not revealed here.
The recommendations of using the NLR on day 2 of ERCP is justified by the AUC and the sensitivity reported in the study. I would look forward to the validation in a larger sample group or as a part of a meta-analysis. Till then the title may reflect that this is only an exploratory study (pilot) to determine the parameters for a larger study of adequate sample size.
My best wishes for future studies by this group.
Author Response
Thank you for this retrospective study on the value of NLR as a predictor of severe PEP.
The centre has performed a large number of ERCPs over 10 years and have a roving average of PEP over 10 years of around 0.4% which is very good (4% over 10 years). These are within the range of international reports.
The statistics have been applied appropriately, though the calculation of power of study is missing. The discussion could mention what would be the appropriate sample size for adequate powering of the study that the authors mention was not met.
A) Thank you very much for your comments. A sample size of 122 patients in each group will be required to achieve a power of 80% with a confidence level of 5% to confirm our results when we assumed a 7.5% incidence rate of mild/moderate PEP and a 0.5% incidence of severe PEP. We have mentioned the calculation of sample size in the discussion section as follows:
“Furthermore, a sample size of 122 patients in each group will be required to achieve a power of 80% with a confidence level of 5% to confirm these results when we assumed a 7.5% incidence rate of mild/moderate PEP and a 0.5% incidence of severe PEP. So the study is underpowered in terms of drawing definitive conclusions on the association between the NLR and severe PEP.”
My primary concern is that there is no statement regarding the ethics of the study, even if it was a retrospective study, it is good practice to at least mention that the ethical concerns were considered e.g. patient confidentiality and consent; even if the medical records are not revealed here.
A) Thank you for the comment. This study was approved by the Institutional Review Board of the hospital. Informed consent was waived by the board. We added a statement regarding the ethics of the study in lines 79-80.
The recommendations of using the NLR on day 2 of ERCP is justified by the AUC and the sensitivity reported in the study. I would look forward to the validation in a larger sample group or as a part of a meta-analysis. Till then the title may reflect that this is only an exploratory study (pilot) to determine the parameters for a larger study of adequate sample size. My best wishes for future studies by this group.
A) We fully agree with your comment that this is only an exploratory study. We will perform future studies for the validation in a larger sample group.
Reviewer 3 Report
This is a well described retrospective analysis and the manuscript is excellently written.
I would like you to add some additional information about what was the standard treatment that was followed at your institution for post ERCP pancreatitis and also include data about rate of intravenous fluid administration/kg after pancreatitis found, how long after ERCP was the diagnosis of pancreatitis made. Also the mild to moderate pancreatitis group had few patients with prophylactic PD stent placement vs severe group had none.
What was the rationale used to decide which patient got PD stent and which patient did not get the stent.
Author Response
This is a well described retrospective analysis and the manuscript is excellently written.
I would like you to add some additional information about what was the standard treatment that was followed at your institution for post ERCP pancreatitis and also include data about rate of intravenous fluid administration/kg after pancreatitis found, how long after ERCP was the diagnosis of pancreatitis made.
A) Thank you very much for your comments. The standard treatment for post-ERCP pancreatitis includes protease inhibitors (e.g., gabexate mesylate, nafamostat, ulinastatin) and aggressive intravenous hydration (3 mL/kg/h for 8 h after the ERCP) with lactated Ringer’s solution. If a patient was suspected of having post-ERCP pancreatitis, administration of protease inhibitors and the rate of fluid therapy remained until 48 h after ERCP. The diagnosis of post-ERCP pancreatitis was made between 6-24 h after ERCP. We described this additional information in lines 67-73 as follows:
“For the prevention of PEP, patients who underwent ERCP received protease inhibitors (e.g., gabexate mesylate, nafamostat, ulinastatin) and aggressive intravenous hydration (a 10-20 mL/kg bolus before the ERCP, 3 mL/kg/h during the ERCP, and 3 mL/kg/h after the ERCP) with lactated Ringer’s solution. The diagnosis of PEP was made between 6-24 h after ERCP. If a patient was suspected of having PEP, administration of protease inhibitors and the rate of fluid therapy remained until 48 h after ERCP. Rectal NSAIDs were not used because those were not available in Korea.”
Also the mild to moderate pancreatitis group had few patients with prophylactic PD stent placement vs severe group had none. What was the rationale used to decide which patient got PD stent and which patient did not get the stent.
A) Thank you for pointing this out. We did not place a pancreatic stent routinely in high-risk patients for post-ERCP pancreatitis. Instead, we placed a pancreatic stent only in case of repeated unintentional cannulation of the pancreatic duct during ERCP. So few patients got the pancreatic stent. We added the indication of a pancreatic stent in our hospital in lines 61-62 as follows:
“A prophylactic pancreatic stent was placed only in case of a repeated unintentional cannulation of the pancreatic duct during ERCP.”